# Bioinformatic Analysis of Roquin Family Reveals Their Potential Role in Immune System

**DOI:** 10.3390/ijms25115859

**Published:** 2024-05-28

**Authors:** Xianpeng Li, Shuaiqi Yang, Xiangmin Zhang, Yi Zhang, Yu Zhang, Hongyan Li

**Affiliations:** 1College of Marine Life Sciences, Institute of Evolution & Marine Biodiversity, Ocean University of China, Qingdao 266003, China; lixianpeng0826@163.com (X.L.); ysqsksg@163.com (S.Y.); 17662131436@163.com (X.Z.); 19861909016@163.com (Y.Z.); 2Laboratory for Marine Biology and Biotechnology, Qingdao Marine Science and Technology Center, Qingdao 266003, China; 3Key Laboratory of Evolution & Marine Biodiversity (Ministry of Education), Ocean University of China, Qingdao 266003, China

**Keywords:** Roquin, evolution, expression, CDE-like motif, immune

## Abstract

The Roquin family is a recognized RNA-binding protein family that plays vital roles in regulating the expression of pro-inflammatory target gene mRNA during the immune process in mammals. However, the evolutionary status of the Roquin family across metazoans remains elusive, and limited studies are found in fish species. In this study, we discovered that the *RC3H* genes underwent a single round of gene duplication from a primitive ancestor during evolution from invertebrates to vertebrates. Furthermore, there were instances of species-specific gene loss events or teleost lineage-specific gene duplications throughout evolution. Domain/motif organization and selective pressure analysis revealed that Roquins exhibit high homology both within members of the family within the same species and across species. The three *rc3h* genes in zebrafish displayed similar expression patterns in early embryos and adult tissues, with *rc3h1b* showing the most prominent expression among them. Additionally, the promoter regions of the zebrafish *rc3h* genes contained numerous transcription factor binding sites similar to those of mammalian homologs. Moreover, the interaction protein network of Roquin and the potential binding motif in the 3’-UTR of putative target genes analysis both indicated that Roquins have the potential to degrade target mRNA through mechanisms similar to those of mammalian homologs. These findings shed light on the evolutionary history of Roquin among metazoans and hypothesized their role in the immune systems of zebrafish.

## 1. Introduction

Precisely regulating gene expression is crucial for all organisms, especially for the development and immune responses of eukaryotes. As a result, abnormal gene expression leads to developmental defects and imbalanced immune responses [1,2,3]. Gene expression is regulated through various mechanisms, such as epigenetic modifications, transcriptional regulation, and post-transcriptional regulation, which also involve multiple regulators. RNA-binding proteins (RBPs) recognize specific RNAs through their RNA-binding domains, thereby regulating RNA splicing, localization, stability, and transport. They play crucial roles in various aspects, including organ development, disease occurrence, and immune system homeostasis [1,4,5,6,7,8,9,10]. Several RBP families, such as TTP, AUF1, KSRP, TIA-1/TIAR, Roquin, Regnase, HuR, and Arid5a, have been identified as important regulators of immune systems at the post-transcriptional level [5]. These proteins typically act as trans-acting factors that bind to specific cis-elements located within the untranslated regions of mRNA. In mammals, Roquin and Regnase are two prominent RBP families known to regulate their targets, thereby controlling immune systems and preventing excessive immune responses.

The structures of the ROQUIN family: The mammalian ROQUIN family includes ROQUIN-1 and a paralog, ROQUIN-2, which are encoded by *RC3H1* and *RC3H2,* respectively [8,11]. *RC3H1* was initially discovered during a screening for autoimmune regulators in mice. It is the causative mutation in the *sanroque* mouse strain and exhibits symptoms characteristic of lupus, an autoimmune disease [11]. ROQUIN proteins are highly conserved and contain multiple domains. Roquin-1 and Roquin-2 share similar domain organizations, including a RING finger domain in the N-terminal, an RNA-binding ROQ domain, and a C3H1 zinc finger domain. The C-terminal sequence features several intrinsically disordered regions, containing a PRR (Proline-Rich Region) with several PxxP, a glutamine/asparagine-rich (Q/N Rich) region, and a CC (coiled-coiled) domain [8,12]. The RING finger domain serves as a distinctive feature of E3 ligase, suggesting that ROQUIN is a potential E3-ubiquitin ligase candidate. The presence of the ROQ domain, a characteristic feature of the Roquin family, has been identified through sequence homology analysis. Subsequent research has confirmed that the ROQ domain is embedded in the HEPN domain, with HEPNN and HEPNC flanking on either side of the ROQ domain. Further studies have involved determining the crystal structure of the ROQ domain in human ROQUIN-1, which has revealed a helical fold bearing a winged helix-turn-helix (wHTH) motif responsible for binding to stem-loop mRNA carrying constitutive decay elements (CDEs) [13]. The C3H1 zinc finger domain is also a common feature of RNA-binding proteins and plays a role in RNA binding [7]. The PRR generally interacts with proteins containing the SH3 domain [8]. The glutamine/asparagine-rich region is responsible for the subcellular localization of Roquin [14]. The CC domain is a prevalent and structurally versatile folding motif with diverse functions based on its specific structure [15].

Roquin is located in P-bodies and regulated by various transcription factors and cytokines. Roquin proteins are predominantly found in the cytoplasm and are enriched in P-bodies, with the ability to translocate to stress granules in response to stress. The ROQ domain plays a key role in localizing Roquin to stress granules [16]. The expression of Roquin family members is regulated by various transcription factors and cytokines. Transcription factors STAT1, STAT3, GATA2, and c-Rel can activate the *RC3H* promoter, whereas IKZF2 represses *RC3H* expression. The immunosuppressive cytokine interleukin IL-10 enhances the activity of these transcription factors, leading to increased *Rc3h1* expression [17]. Studies of *Il10*^−/−^ mice have shown reduced Roquin-1, further indicating that IL-10 plays a role in regulating the expression of *Rc3h1* [18].

The role of the ROQUIN family in immunity: Roquin-1 and Roquin-2 have important and redundant roles in both innate and adaptive immunity. They regulate common mRNA targets, such as ICOS, Ox40, IFN-γ and TNF-α, but also have many distinct targets. They bind to the 3′-UTR region of target mRNA, destabilizing the mRNAs and, thus, preventing immune cell over-activation. ICOS, a costimulatory receptor for follicular T helper cells, is the first identified target of Roquin-1 [11]. Roquin promotes the ICOS degradation. In *sanroque* mice, ICOS is overexpressed, leading to aberrant Tfh cell accumulation. Roquin inhibits TH17 cell differentiation, leading to inhibition of target mRNA encoding the TH17 cell-promoting factors IL-6, ICOS, c-Rel and IRF4. Upon activating T cell receptor and co-stimulatory signaling, MALT1 cleaves and inactivates Roquin-1 and Roquin-2, thereby enhancing TH17 cell differentiation and induced humoral autoimmunity and organismal defense [19,20,21]. The first identified sequence recognized by Roquin-1 is a conserved class of stem-loop RNA degradation motifs located in the 3′-UTR region known as the CRE (Conserved Cis-Regulatory Element). This motif contains an AU-rich consensus sequence, 5′-NNNNNUUCYRYGAANNNNN-3′. Roquin-1 directly binds to this element and represses the expression of target mRNA involved in innate and adaptive immunity. In addition to CRE, an alternative decay element (ADE), a U-rich hexaloop motif, has been identified using the SELEX assay. This ADE and the previously identified CDE cooperate in the repression of Ox40 via Roquin [22]. Indeed, the mechanism of Roquin regulation is quite complex. Roquin targets mRNAs in their 3′-UTR regions through diverse modes of regulation that could interact with SL structures as well as linear sequence elements [23]. Most targets are degraded by mRNA decay; however, a small subset also experiences translational inhibition [23]. The recognition of Roquin to the target is mediated by the ROQ domain, which has two separate RNA-binding sites (A and B), the A site being for stem-loop RNA and the B site being for double-stranded RNA [24].

In addition to mammalian Roquin, there have been several studies about Roquin in other species. The *C. elegans* homolog of *RC3H1*, RLE-1 (regulation of longevity by E3), was initially identified as an E3 ubiquitin ligase of transcription factor DAF-16, regulating aging [25]. Later, it was proven that RLE-1 post-transcriptionally regulates ETS-4, which is the master transcriptional regulator of diverse effectors [26]. The *Drosophila melanogaster* homolog of Roquin recruits the CCR4-Not complex through a CAF40-binding motif and represses the expression of its target mRNAs [27]. Later, it was found that Roquin regulates the STING-dependent immune response negatively in *Drosophila* [28]. However, the phylogenetic evolution of the Roquin family remains unclear, and its homologs remain undermined in teleost species. In this study, we explored the phylogenetic evolution of the Roquin family in metazoans; then, we examined the expression profiles and regulation of Roquins in zebrafish.

## 2. Results

### 2.1. Roquin Genes in Metazoan

Taking advantage of extensive genomic data, we acquired *RC3H* sequences of various animal species at different evolutionary positions from the NCBI and Ensembl databases, using human *RC3H1* as a query. We observed the presence of *RC3H* genes in all the species chosen, although the exact numbers varied between species (Table 1). Notably, both *RC3H1* and *RC3H2* were identified in humans, mice, and chickens. It is worth mentioning that *Rc3h2* was not detected in *Xenopus tropicalis*; thus, we searched amphibian species, and interestingly, we found that only one *Rc3h1* was present in all *Anura* species, excluding *Xenopus laevis*, which is a tetraploid and contains two duplicated *Rc3h1s*. However, both *Rc3h1* and *Rc3h2* are present in *Gymnophiona* species. These data suggest that *Rc3h2* is specifically lost in Anura species. In teleost species, such as the spotted gar and torafugu, only two members (*rc3h1* and *rc3h2*) have been identified. Rainbow trout exhibit six members, consisting of four copies of *rc3h1* (*rc3h1aa*, *rc3h1ab*, *rc3h1ba* and *rc3h1bb*) and two copies of *rc3h2* (*rc3h2a* and *rc3h2b*). Zebrafish have three family members, *rc3h1a*, *rc3h1b* and *rc3h2*. Most cartilage fishes had both *rc3h1* and *rc3h2*. In invertebrates, only one member, *rc3h1*, is represented (the homologous gene in nematodes is referrd to as *rle-1*).

We conducted an analysis of Roquins in zebrafish. Encoded Roquin-1a consisted of 1078 amino acids, with a molecular weight of approximately 120.12 kDa and an isoelectric point (pI) of around 7.87. Roquin-1b was similar to Roquin-1a, comprising 1111 amino acids, with a molecular weight of approximately 122.61 kDa and a pI of approximately 7.41. Roquin-2 was slightly shorter than Roquin-1a and Roquin-1b, containing 1028 amino acids, with a molecular weight of approximately 113.96 kDa and a pI of around 6.56 (Table 2).

### 2.2. Evolutionary Relationship of ROQUINS

To gain a deeper understanding of the evolutionary relationships of Roquin homologs in metazoans, we meticulously aligned the retrieved sequences of the Roquin proteins using the ClustalW algorithm. Additionally, we comprehensively analyzed their phylogenetic evolution by constructing two phylogenetic trees using Neighbor-Joining (NJ) and Maximum-Likelihood (ML) methods, available in MEGA10. Remarkably, the resulting trees generated from both methods were generally consistent (Figure 1A,B). In vertebrates, Roquin-1 and Roquin-2 were found to belong to two distinct clades, indicating their evolutionary divergence. Interestingly, all the invertebrate Roquins were located at the base of the vertebrate clades. This suggests that ROQUIN-1 and ROQUIN-2 originated from a single, primitive Roquin-1 in invertebrates. Notably, in teleost species, Roquin-1 was divided into two branches, namely, Roquin-1a and Roquin-1b. These observations suggest that these teleost-specific branches originated from extra genome duplication in teleost. In rainbow trout, Roquin-1aa and 1ab; Roquin-1ba and 1bb; and Roquin-2a and 2b clustered together within distinct sub-branches. This suggests that these homologs are likely derived from a fourth whole genome duplications (WGDs) event in rainbow trout.

### 2.3. Genomic Structure and Synteny of Rc3h Genes

We conducted an analysis and created a visual representation of the genomic structure of Rc3h genes in humans, mice, and zebrafish. The genomic structure of Rc3hs was conserved among genes and across different species (Figure 2). Interestingly, the exon–intron organization of Rc3h1 genes was generally conserved in each species, consisting of 20 exons and 19 introns. The only exception was rc3h1b, which had only 19 exons and 18 introns. The exon composition of Rc3h2 varied between the three species. There were 19 exons in zebrafish, 22 exons in mice, and 21 in humans. In addition, it is noteworthy that the length of these genes varied, with shorter rc3h genes in zebrafish (Figure 2).

We also analyzed the synteny of Rc3hs and found no synteny between the invertebrate rc3h1 genes and the vertebrate Rc3h genes. Additionally, we illustrated a synteny map of rc3h genes in zebrafish and humans (Figure 3). The results revealed that rc3h1a in zebrafish exhibited a well-conserved collinearity with RC3H1 in humans. Downstream of rc3h1a and RC3H1, two neighboring genes, serpinc1 and zbtb37, were identified. However, in zebrafish rc3h1b, the genomic region underwent significant rearrangement, and no copies of these two neighboring genes were found. Combined with phylogenetic analysis, these finding suggest that rc3h1a and rc3h1b are orthologs of RC3H1 and originated from a third WGD event in zebrafish. For rc3h2, two neighboring genes, strbp and rabgap1, were identified in both zebrafish and humans. Although these two genes share the same orientation, their location differs between zebrafish and humans, indicating a reorganization of the neighboring genes.

### 2.4. Domain and Motif Organization of Roquins

To better understand the functional diversification of Roquins, we analyzed the domains and motifs of Roquins of invertebrate amphioxi, vertebrate zebrafish, and humans using the MEME and SMART protein domain prediction program combined with Megalign analysis. The results revealed that all Roquin possess a Ring finger domain, a ROQ domain, and a C3H1-ZNF domain located at the N-terminal (Figure 4). The C-terminal sequence of all the Roquins exhibited prominent multiple intrinsically disordered fragments. These fragments contained a PRR with several PxxPs, as well as a glutamine/asparagine-rich (Q/N Rich) region, although the lengths of these regions varied between paralogs and across species. Additionally, a CC (coiled-coiled) domain was present in all Roquins except for Roquin-2 in zebrafish, although the length of the CC domain differed between paralogs and across species (Figure 4). These results indicate that the domains of vertebrate ROQUIN proteins are highly conserved, implying that their functions are also somewhat conserved.

We identified ten motifs and labeled them motifs 1–10 based on their level of conservation, with motif 1 being the most conserved (Figure 5). The results revealed that all Roquin proteins contained these ten motifs and exhibited similar motif organization. The N-terminal to C-terminal motif organization was as follows: motif 1, 10, 7, 3, 5, 2, 4, 8, 6, and 9, with respective amino acid lengths of 50, 30, 27, 50, 29, 50, 50, 29, 25, and 42. The nine motifs located at the N-terminal corresponded to the RING finger domain, the ROQ domain, and the C3H1-Znf domain. Notably, these nine motifs were clustered together and relatively conserved, while the last motif at the C-terminal was dispersed and not conserved in its position (Figure 5). The first motif, located at the N-terminal of the sequence, exhibited the highest degree of conservation and spanned 50 amino acids, resembling the RING finger domain found in Roquin-1. Motifs 7, 3, 5, 2, 4, and 8 collectively formed the ROQ domain, while motif 6 contained the C3H1-Znf domain. The conserved domain/motif organization of these Roquins suggests that they have similar functions.

### 2.5. Selective Pressure of rc3h Genes

To investigate whether the duplicated Roquin genes underwent selection pressures, we calculated nonsynonymous (Ka), synonymous (Ks), and Ka/Ks ratios for the *rc3h* gene pairs in vertebrate zebrafish and invertebrate amphioxi.

The Ka/Ks ratio is a measure of selective pressure on protein-coding genes. The results demonstrated that the Ka/Ks ratios of all the *rc3h* gene pairs ranged from 0.1089 to 0.2738. Interestingly, the *rc3h* gene pairs in zebrafish exhibited slightly lower ratios compared with those between different species. Furthermore, all the Ka/Ks ratios were much lower than 0.5, indicating that they had all experienced purifying selection and lower evolutionary pressure (Table 3). In summary, the above analysis suggests that Roquin-1 and Roquin-2 are evolutionarily conserved and that their functions may be redundant to some extent.

### 2.6. Expression Profile of rc3h Genes in Zebrafish

We analyzed the expression profile of rc3h genes in the early development and adult tissues of zebrafish. All three genes exhibited similar expression patterns in the early embryos, with particular emphasis on the fact that rc3h1b had the strongest expression level among the three. Rc3h1a showed modest expression both maternally and zygotically throughout all zebrafish development stages. Rc3h1b exhibited robust expression in all the tested stages, with the highest expression level observed from 2.25 h post-fertilization (hpf) to 8 hpf. On the other hand, rc3h2 only showed weak expression during embryonic development (Figure 6A). Collectively, these data suggest that Rc3h genes are involved in the early development of zebrafish, with a key role that can be attributed to Rc3h1b.

The expression profile of rc3h genes in adult tissues revealed similar expression patterns with slight differences. Rc3h1a exhibited modest expression in all tissues, with the highest expression in the testes, followed by the skin, brain, and kidneys. Rc3h1b showed the strongest expression among the three genes, with the highest levels observed in the testes, followed by the brain, kidneys, and intestines. Rc3h2 displayed weak expression compared with rc3h1a and rc3h1b, with the highest expression in the brain, followed by the intestines, skin, and kidneys (Figure 6B). These data indicate that the function of Roquin is redundant, with slight divergence.

### 2.7. Potential Transcription Factor in the Promoters of rc3h Genes

Transcription factors (TFs) Stat1, Stat3, Gata2, and c-Rel can upregulate Roquin expression, while Ikzf2 can downregulate it. Therefore, we predicted potential TF-binding sites within the 2 kb promoter region of the *rc3h* genes. The analysis revealed the presence of abundant TF-binding sites in the promoters of *rc3h* genes in both zebrafish and amphioxi (Figure 7). Interestingly, some of these TF binding sites overlapped. In detail, in the zebrafish *rc3h1a* gene, five Stat1β-binding sites, seven Gata2-binding sites, and twenty-three Ikzf2-binding sites were identified. In zebrafish *rc3h1b*, all five TF-binding sites were found, including one Stat3, eleven c-Rel, two Stat1β, one Ikzf2, and twelve Gata2. Zebrafish *rc3h2* contained TF-binding sites, including three Gata2, seven Ikzf2, and nineteen Stat1β. In amphioxi *rc3h1*, there were fifteen Ikzf2-, nineteen Gata2-, one c-Rel-, and three Stat1β-binding sites. These findings highlight the abundance and potential functional importance of these TF-binding sites in regulating *rc3h* genes. Despite variations in the location and number of each TF-binding site across different genes, it can be inferred that there is a conserved pattern in the *rc3h* gene regulated by TFs between paralogs and across species.

### 2.8. Interaction Protein Network and Potential Binding Motif of Roquins in Zebrafish

According to the STRING protein–protein interaction database, zebrafish Roquin-1a can interact with Cnot1 and Cnot9, both of which are members of the CCR4-NOT complex. Roquin-1a can also interact with Roquin-2 and shows co-expression with Zc3h12a and Roquin-2 (Figure 8A). Zebrafish Roquin-1b, on the other hand, interacts with Cnot1, Cnot9, Cnot11 and Cnot3a, all of which are components of the CCR4-NOT complex. Additionally, Roquin-1b also exhibits co-expression and interaction with Roquin-2 (Figure 8B). Zebrafish Roquin-2 can interact with Cnot1, a member of the CCR4-NOT complex. Interestingly, Roquin-2 can bind to Helz and shows co-expression with Arid5a (Figure 8C), both of which are involved in immune hemostasis post-transcriptionally. These data suggest that all Roquins have the potential to promote mRNA target decay by recruiting the CCR4-NOT complex.

It has been confirmed that Roquin degrades target mRNA by binding to the constitutive decay element (CDE) in the 3’-UTR region of mRNA, thereby limiting the accumulation of harmful inflammatory factors. Through NCBI alignment, we identified a putative 13-nucleotide CDE-like motif in zebrafish, which is shorter than the 17 nt CDE found in mammals (Figure 9A,B). We found this CDE-like motif in at least 12 genes in zebrafish, most of which are closely related to immune and inflammatory processes, such as *tnfα*, *smarca2*, *prkca*, and *stk10*. These findings further indicate that the mechanism of mRNA degradation caused by ROQUIN is highly conserved across vertebrates.

## 3. Discussion

The Regnase and Roquin RNA-binding proteins collaboratively control the degradation of mRNA and maintain homeostasis and the immune system. Previously, we elucidated the evolution of the Regnase-encoded *Zc3h12* gene family and uncovered the immunomodulatory role of Regnases in zebrafish [29]. However, information regarding the evolution of Roquins in zebrafish is lacking. In this study, we aimed to explore the evolutionary relationships of the Roquin family in metazoans and examined its expression profiles, regulation, and potential role in zebrafish.

Genome duplication increased gene numbers, thus leading to an increase in gene numbers and an expansion of gene families, which is important for genome evolution and genetic robustness [30,31,32,33,34]. We observed that invertebrates and jawless vertebrates possess one *Rc3h* gene, whereas mammals, owing to WGD, have two *Rc3h* genes (*Rc3h1* and *Rc3h2*). Interestingly, in the *Anura* amphibian, the ortholog of *Rc3h2* has been lost. In teleost, which experienced third and fourth WGD events, the number of *rc3h* genes varies from two to six. These findings suggest that during evolution from invertebrates to jawless vertebrate lamprey to vertebrates, only one round of *rc3h* gene duplication occurred. However, in teleost, additional gene duplication events took place. Moreover, there were species-specific gene loss events throughout this evolution. Across invertebrates, jawless vertebrates, and vertebrates, there are generally one, two, and four *Zc3h12* genes, respectively. Notably, in bony fish, which experienced three or four rounds of WGD events, the number of *zc3h12* genes varies greatly [29]. *Zc3h12* underwent two rounds of gene duplication events, along with exceptional species-specific gene duplication or loss events during teleost evolution [29]. Both the Roquin and Regnase families participate in post-transcriptionally regulating mRNA and are involved in immune responses. The *rc3h* and *zc3h12* gene families in metazoans were generated through WGD and lineage-specific gene duplication/loss events. However, the *rc3h* gene family experienced one less round of duplication compared with *zc3h12* during evolution. As a result, *rc3h* gene family members are much simpler across species compared with *zc3h12*.

Roquins share significant homology and generally exhibit a high level of conservation within both family members within the same species and across species, although there may be slightly fewer variations in domain and motif composition. Our analysis of Ka/Ks ratios for the *rc3h* gene pairs of vertebrate zebrafish and invertebrate amphioxi indicates that purifying selection and lower evolutionary pressure have influenced their evolution. The putative TF binding site in the promoters of invertebrate amphioxi *rc3h1* and teleost zebrafish *rc3h* genes suggests that *rc3h* genes are regulated by similar TFs, which is consistent with previous studies on mice [17]. These bioinformatic data suggest that Roquins perform similar functions across species and exhibit redundant roles among different members within the same species. Roquins in vertebrate humans, invertebrate flies and nematodes exhibit similar subcellular localization: they localize mainly in P-bodies in the cytoplasm, concentrate in stress granules in response to stress, function as RNA-binding proteins, and trigger mRNA decay [16]. In mice, both Roquin-1 and Roquin-2 can bind nucleic acids and demonstrate functional redundancy [35,36]. Roquins and Regnases can cooperate or act independently to regulate mRNA silencing post-transcriptionally [19,37]. Nematode Rle-1 (Roquin-1) has conserved functions, and both Rle-1 and REGE-1 work collaboratively yet independently to regulate ets-4 mRNA silencing [26]. *Drosophila* Roquin regulates innate immune responses by inhibiting STING signaling [28]. *Rc3h1b* expression is most prominent among the three paralogs in zebrafish, suggesting that Roquin-1b is the primary player in zebrafish.

Our analysis of domain and motif organization implies that the C-terminal regions of Roquin show variability compared with the highly conserved N-terminal region, which is consistent with a previous report [38]. In humans, zebrafish, and amphioxi, only one motif consisting of 42 amino acids was identified in the C-terminal region of Roquin, but its function is not yet defined. Despite the low similarity in the C-terminal region, Roquins exhibit multiple intrinsically disordered fragments, indicating a conserved function. Human Roquins are known to promote RNA degradation by recruiting the CCR4-NOT deadenylase complex, thus preventing autoimmunity [39]. Interestingly, *Drosophila* Roquin can also interact with the CCR4-NOT complex and mediate the degradation of bound mRNA targets [27]. It has been demonstrated that the Roquin C-terminal region is responsible for interaction with the CCR4-NOT complex [27]. Consistently, the protein–protein interaction network of Roquins suggests that all zebrafish Roquins are capable of interacting with the CCR4-Not complex, which mediates mRNA target decay. Notably, zebrafish Roquin-2 can bind Helz, which is an interaction partner of the CCR4-NOT complex and acts as a mediator of mRNA decay [40]. Furthermore, zebrafish Roquin-2 exhibits co-expression with Arid5a, which is also a well-known RNA-binding protein involved in multiple immune pathways. Arid5a is known to regulate several IL-17 mRNA targets by promoting their stability and/or translation [41]. The mammalian ROQUIN degrades target-gene mRNA by binding to the CDE motif [38]. In our study, we identified a similar CDE motif in many immune/inflammatory genes of zebrafish. Therefore, it can be speculated that the roles and mechanisms of Roquins are conserved across vertebrates.

This study investigated the evolutionary history of *rc3h* genes among metazoans. Additionally, we analyzed the expression profiles and regulation of *rc3h* genes in zebrafish. Furthermore, we hypothesized that zebrafish Roquins play a role in post-transcriptional immune responses. Our findings provide insights into the evolution of *rc3h* and its potential involvement in the immune systems of Roquin in zebrafish.

## 4. Materials and Methods

### 4.1. Sequence Retrieval and characterization

To obtain Roquin family genes in selected species, we conducted a search on GenBank (http://www.ncbi.nlm.nih.gov, accessed on 18 May 2024) and Ensembl (http://www.ensembl.org, accessed on 18 May 2024) using *homo sapiens RC3H1* as the query sequence by BLAST. Redundant transcript sequences of the same gene were removed, and the candidate *RC3H* genes were further verified by predicting the conserved domain of the encoded proteins using the Smart program v9.0 (http://smart.embl-heidelberg.de, accessed on 28 April 2024) combined with Megalign analysis. Then, the domain organization of Roquins were drawn with IBS v1.0 software (Changsha, China. http://ibs.biocuckoo.org/, accessed on 5 May 2024).

Subsequently, we analyzed the characteristics of these proteins. Specifically, the molecular weight and isoelectric point (pI) of the Roquin proteins were calculated with the online pI/Mw tool v3.0 (https://web.expasy.org/compute_pi, accessed on 16 April 2024). Sequence similarity and divergence were aligned using Megalign.

We further analyzed the protein motifs of Roquin with Multiple Em for the Motif Elicitation (MEME) tool v5.5.5 [42] (https://meme-suite.org/meme/tools/meme, accessed on 11 May 2024) in the neighborhood of homology, selected the “Classic mode” option and the “DNA, RNA or Protein” option, entered the amino acid sequences to be analyzed, and set the number of expected motifs to 10.

### 4.2. Phylogenetic Analysis

Using human and mouse ROQUIN-1 and ROQUIN-2 protein sequences, a BLAST of ROQUIN-1 and ROQUIN-2 protein homologs from other species was performed using GenBank and Ensembl. Then, we removed the redundant sequences of the same proteins and checked them one by one to obtain all ROQUIN-1 and ROQUIN-2 protein sequences.

We performed a multiple alignment of the Roquin proteins from all the selected species. To construct a phylogenetic tree, we utilized both the Maximum-Likelihood and Neighbor-Joining methods of MEGA11. The tree was generated with a bootstrap of 1000 replicates, and the evolutionary distances were computed using the JTT matrix-based method. We used Figtree v1.4.4 to modify the phylogenetic trees.

### 4.3. Genomic Structure, Synteny Analysis and Selective Pressure Analysis

The genomic intron–exon structures of *rc3h* genes were generated using Visualize Gene Structure of TBtools v2.056 (Guangzhou, China. https://github.com/CJ-Chen/TBtool, accessed on 9 April 2024) [43], and the genomic structure diagram was drawn with IBS software. All the genomic data were downloaded from the NCBI Reference Sequence Database (ftp://ftp.ncbi.nlm.nih.gov/genomes/refseq/, accessed on 18 May 2024).

The gene syntenic data of *rc3h* genes of the zebrafish and human were collected from Genome Date Viewer (https://www.ncbi.nlm.nih.gov/genome/gdv, accessed on 15 May 2024) and Genomicus v100.01 (https://www.genomicus.bio.ens.psl.eu/genomicus-100.01/cgi-bin/search.pl, accessed on 20 April 2024) [44]. The figure was generated using the IBS software v1.0. The Ka/Ks ratios were generated using Simple Ka/Ks Calculator (NG) of TBtools v2.056.

### 4.4. Expression Analysis of rc3h Genes by Online Data

The mRNA expression profiles of zebrafish *rc3h* genes during different stage of embryonic development and in various adult tissues were retrieved from online data [45,46].

### 4.5. Putative Transcription Factors Binding Sites

The 2 kb promoter sequences of *rc3h1* and *rc3h2* were retrieved from the NCBI database. The PROMO database (https://alggen.lsi.upc.es/, accessed on 3 May 2024) was utilized to identify the putative TF binding sites [47], with a maximum matrix dissimilarity rate set to less than 15. Using Adobe Illustrator 2020, visual representations were created based on the location of the predicted transcription factor binding sites.

### 4.6. Analysis of Protein Interaction Network and Potential Binding Motif of Roquins in Zebrafish

The protein–protein interactions (PPI) of Roquin were analyzed using the STRING database (https://cn.string-db.org/, accessed on 21 April 2024).

Using the UGUUUUCUGUGAAAACA motif of *TNFα* in mammals as a reference, we performed a sequence BLAST in the zebrafish genome to identify potential CDE motifs. Nucleotide BLAST was selected, Query Sequence was entered, and Reference RNA sequences (refseq rna) was selected. Subsequently, we retrieved candidate mRNAs containing this motif.

## Figures and Tables

**Figure 1 ijms-25-05859-f001:**
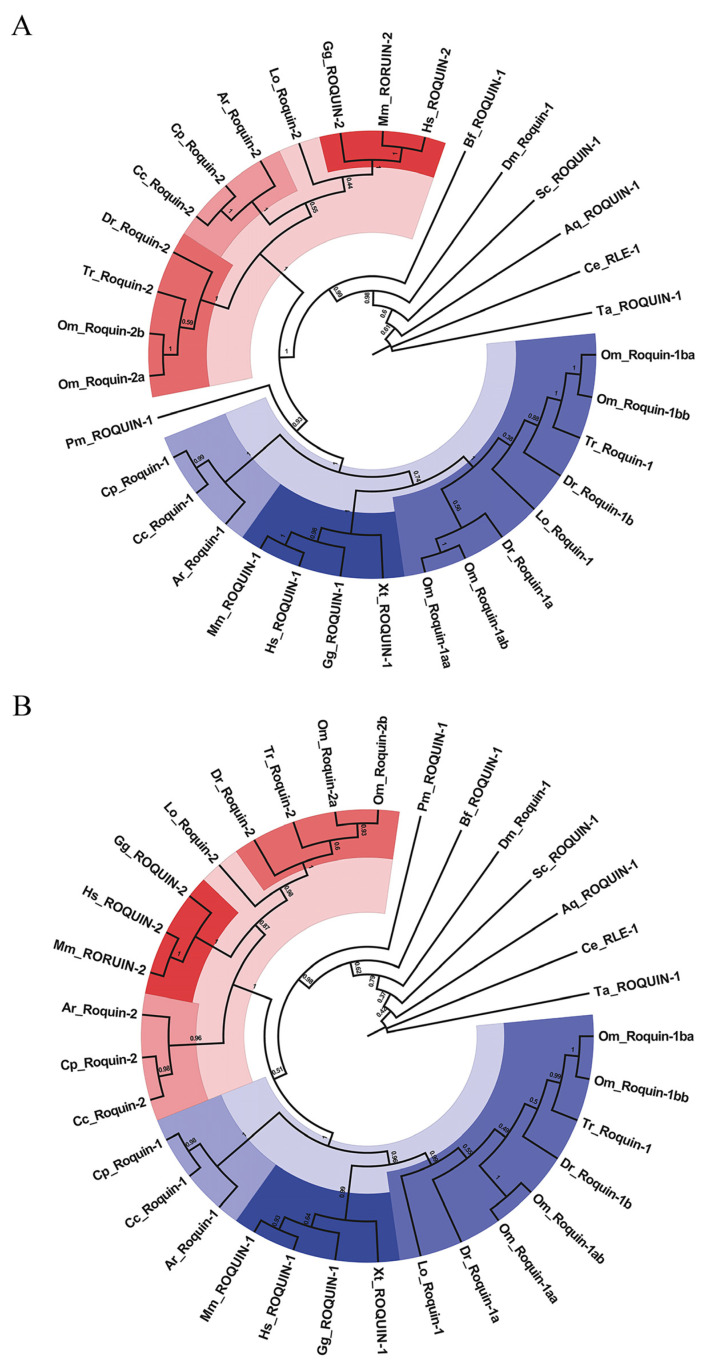
The phylogenetic trees of Roquins between various species constructed by MEGA with two different methods: (**A**) Neighbor-Joining; and (**B**) Maximum-Likelihood. Homologous proteins within the same branch are represented by the same color, with darker shades indicating closer evolutionary relationships. Bootstrap values (%), displayed at each node after 1000 replicates, depict the confidence levels of the branching patterns. Hs, Homo sapiens; Mm, Mus musculus; Gg, Gallus gallus; Xt, Xenopus tropicalis; Dr, Danio rerio; Lo, Lepisosteus oculatus; Om, Oncorhynchus mykiss; Tr, Takifugu rubripes; Cc, Carcharodon Carcharias; Cp, Chiloscyllium plagiosum; Ar, Amblyraja radiate; Pm, Petromyzon marinus; Bf, Branchiostoma floridae; Sc, Styela clava; Dm, Drosophila melanogaster; Aq, Amphimedon queenslandica; Ce, Caenorhabditis elegans; Ta, Trichoplax adhaerens.

**Figure 2 ijms-25-05859-f002:**
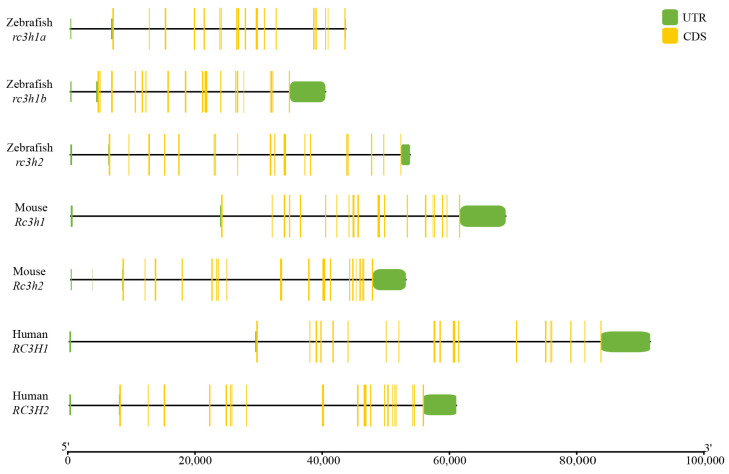
The exon–intron structure of the *RC3H1* and *RC3H2* genes between different species. Rectangles represent exons, while lines signify introns; green rectangles denote Untranslated Regions; yellow rectangles indicate Coding Sequences. The scale bar at the bottom provides a reference for gene length.

**Figure 3 ijms-25-05859-f003:**
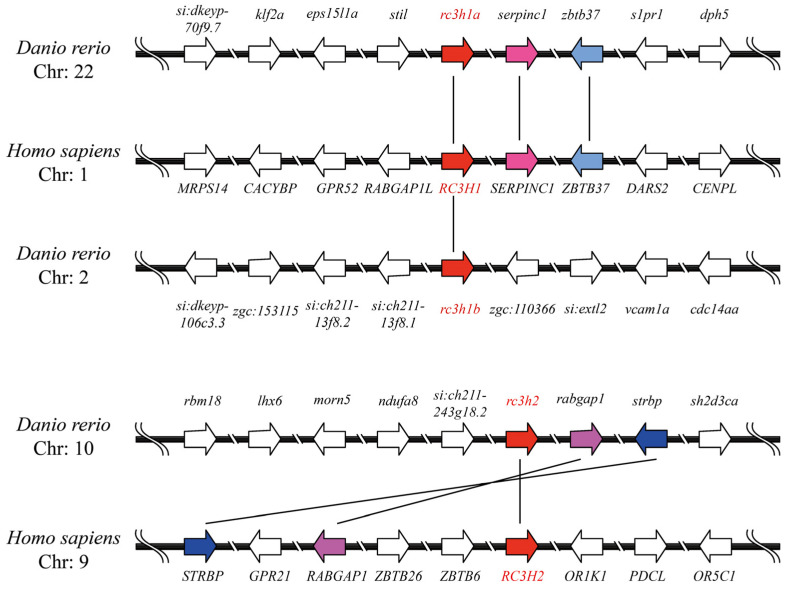
The collinearity analysis of the *RC3H1* and *RC3H2* genes between humans and zebrafish. All genes are represented by arrows, with the direction of the arrow indicating the gene’s orientation. The same color of arrows suggests homologous genes between different species. Chr, Chromosome.

**Figure 4 ijms-25-05859-f004:**
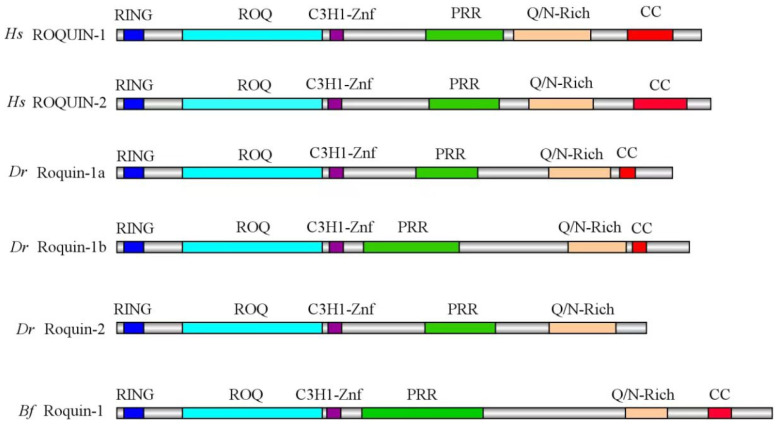
The schematic diagrams of the domain composition of ROQUIN-1 and ROQUIN-2 proteins between various species. RING, ROQ, C3H1-Znf, PRR, Q/N-Rich, and CC domains are shown in colored boxes on a grey background, representing the full length of the proteins. Each domain is depicted with a unique color for visualization. Hs, Homo sapiens; Dr, Danio rerio; Bf, Branchiostoma floridae.

**Figure 5 ijms-25-05859-f005:**
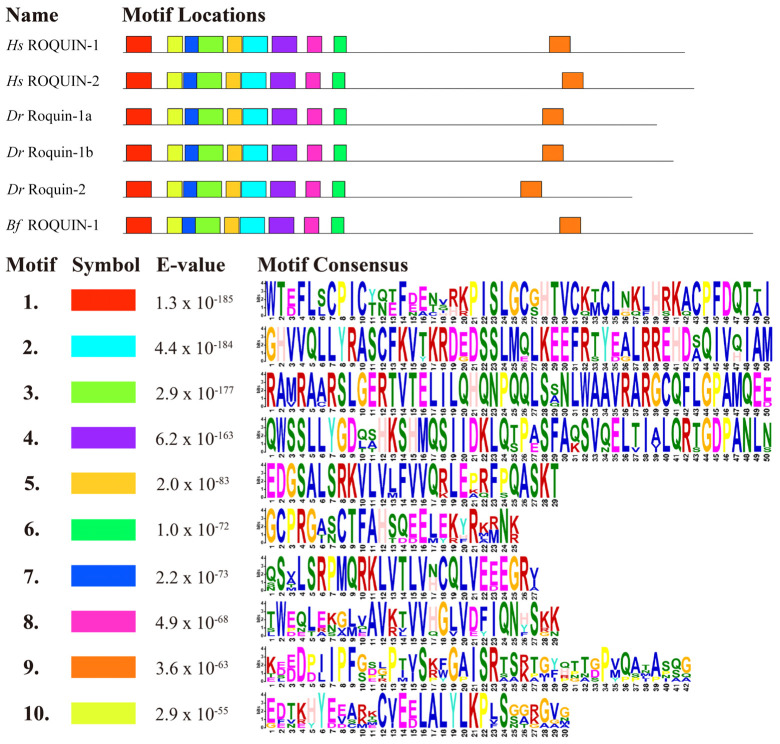
Analysis of the protein motifs of ROQUIN-1 and ROQUIN-2 between various species. The motifs are illustrated in colored boxes. The letters within each motif stand for the abbreviations of amino acids. Larger letters signify higher conservation, indicating a greater probability of the amino acid appearing at the same position within the motif between various species. Hs, Homo sapiens; Dr, Danio rerio; Bf, Branchiostoma floridae.

**Figure 6 ijms-25-05859-f006:**
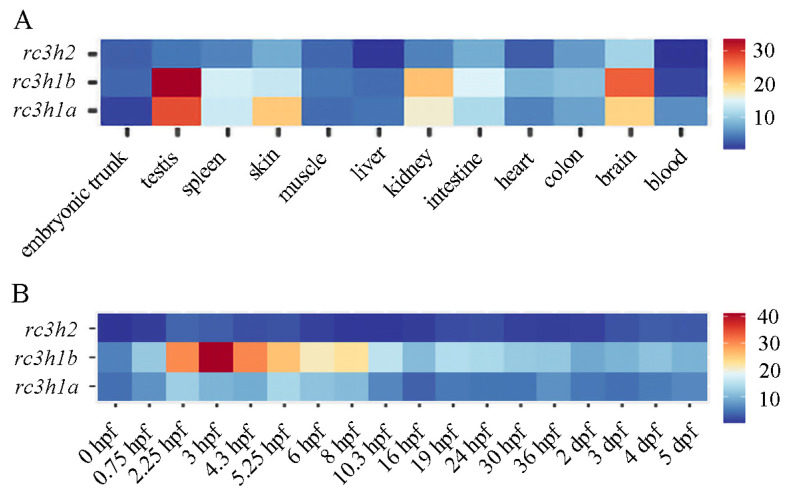
The heatmap displays the expression levels of zebrafish *rc3h1a*, *rc3h1b*, and *rc3h2* genes between different stages and tissues. All expression levels between different stages (**A**) and tissues (**B**) are derived from RNA-seq data. The colors range from dark blue to dark red, reflecting low expression levels to high expression levels.

**Figure 7 ijms-25-05859-f007:**
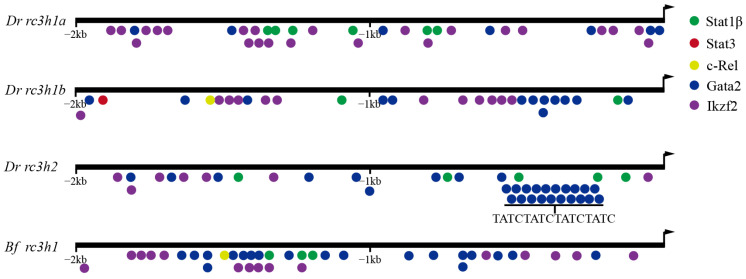
Analysis of transcription factor binding sites in the promoter regions of the *rc3h1* and *rc3h2* genes in zebrafish and amphioxi. The two kb promoter regions are shown, with arrows indicating the transcription start site. Different-colored dots represent the binding sites of different transcription factors. Note that there are sequences in the promoter of the zebrafish *rc3h2* gene that contain multiple overlapped binding sites for Gata2.

**Figure 8 ijms-25-05859-f008:**
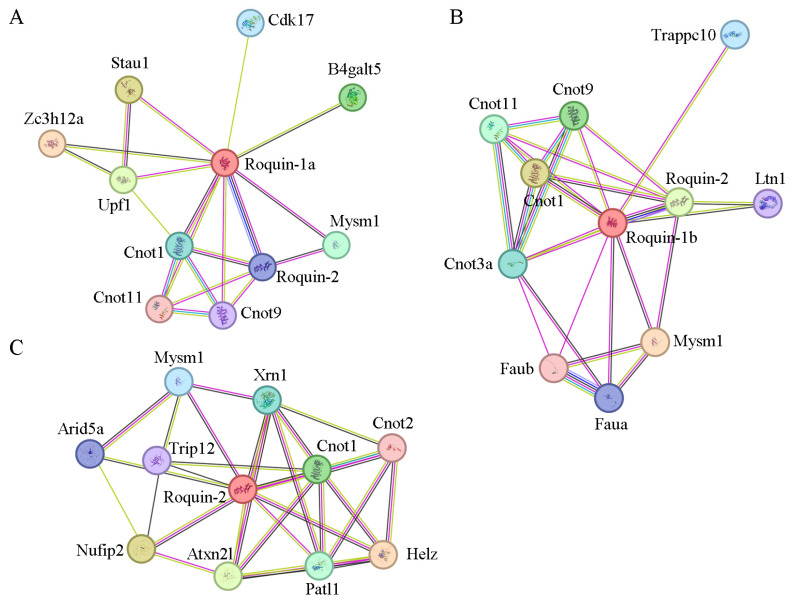
Protein–protein interaction network analysis of Roquin-1a, Roquin-1b, and Roquin-2 in zebrafish. (**A**) Protein–protein interaction network analysis of Roquin-1a. (**B**) Protein–protein interaction network analysis of Roquin-1b. (**C**) Protein–protein interaction network analysis of Roquin-2. Network nodes represent proteins.

**Figure 9 ijms-25-05859-f009:**
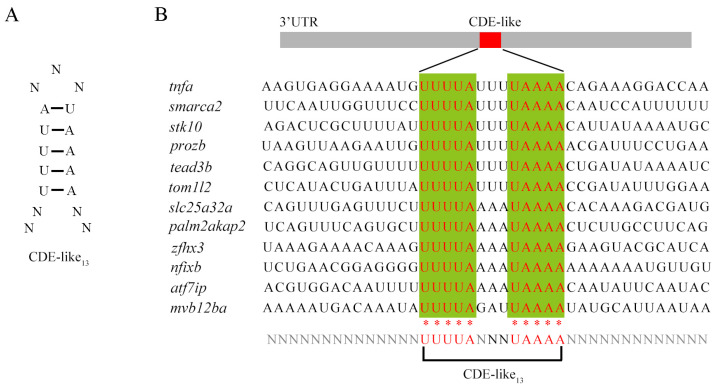
The schematic diagram shows the 3’-UTR region of multiple genes containing CDE-like motifs. (**A**) Secondary structure model of CDE-like motif. (**B**) Genes containing the CDE-like motif identified in zebrafish; this motif presents in the 3’-UTR region of mRNAs. Red nucleotides represent 100% conservation, marked with * at the bottom. N refers as an undefined nucleotide.

**Table 1 ijms-25-05859-t001:** The numbers of *RC3H* genes among different species.

Class	Species	*Rc3h1*	*Rc3h2*	Total
Mammalia	Human	1	1	2
	Mouse	1	1	2
Aves	Chicken	1	1	2
Amphibia	Frog	1	0	1
Osteichthyes	Spotted gar	1	1	2
	Rainbow trout	4 (1aa, 1ab, 1ba, 1bb)	2 (2a, 2b)	6
	Torafugu	1	1	2
	Zebrafish	2 (1a, 1b)	1	3
Chondrichthyes ^1^	Shark	1	1	2
	Ray	1	1	2
Cyclostomata	Lamprey	1		1
Leptocardii	Amphioxi	1		1
Ascidiacea	Ascidian	1		1
Insecta	Fly	1		1
Phasmidia	Nematode	1		1
Demospongiae	Sponge	1		1
Trichoplacoidea	Trichoplax	1		1

^1^ The Chondrichthyes include the whiteshark (*Carcharodon carcharias*), the bambooshark (*Chiloscyllium plagiosum*), and the skate (*Amblyraja radiata*).

**Table 2 ijms-25-05859-t002:** Summary of characteristics of *rc3h* genes in zebrafish.

Gene	Chromosome	No. Exon	CDS (bp)	Amino Acids (aa)	Molecular Weight (Da)	pI
*rc3h1a*	22	20	3237	1078	120125.74	7.87
*rc3h1b*	2	19	3336	1111	122613.65	7.41
*rc3h2*	10	19	3087	1028	113962.46	6.56

Abbreviations: CDS, Coding Sequence; pI, isoelectric point.

**Table 3 ijms-25-05859-t003:** The Ka/Ks ratios of the *rc3h1* and *rc3h2* genes in zebrafish and amphioxi.

Seq_1	Seq_2	Ka	Ks	Ka/Ks
Bf-*rc3h1*	Dr-*rc3h1a*	0.494146491	1.804421487	0.27385314
Bf-*rc3h1*	Dr-*rc3h1b*	0.445397577	2.009173201	0.221682022
Bf-*rc3h1*	Dr-*rc3h2*	0.492221298	2.375139538	0.207238897
Dr-*rc3h1a*	Dr-*rc3h1b*	0.221189798	2.031204763	0.108895864
Dr-*rc3h1a*	Dr-*rc3h2*	0.428709628	2.433626247	0.176160834
Dr-*rc3h1b*	Dr-*rc3h2*	0.416147921	1.995249095	0.208569407

Abbreviations: Dr, Danio rerio; Bf, Branchiostoma floridae. Ka: nonsynonymous substitution rate; Ks: synonymous substitution rate.

## Data Availability

The transcript sequences and protein sequences of all species required for the research analysis in this paper were searched and downloaded from NCBI.

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
