# Peer review of "Bioinformatic Analysis of Roquin Family Reveals Their Potential Role in Immune System"

_ijms, 2024, doi:10.3390/ijms25115859_

Round 1

Reviewer 1 Report

Comments and Suggestions for Authors

Introduction:

·       The presentation of the introduction is not precise or well-defined. It lacks a clear transition between going into specifics of RNA binding proteins (RBPs) and the broader significance of controlling gene expression.

·       Certain statements, such "Abnormal gene expression leads to developmental defects and imbalanced immune responses," are not properly cited or supported by data. Although this is generally accurate, it seems like an unsupported claim in the absence of particular references.

·       It is overly generalised and simplistic to say that "RNA binding proteins play a crucial role in post-transcriptional control of gene expression, impacting almost all aspects of RNA metabolism". It ignores the variety of roles played by various RNA-binding proteins as well as the intricacy of post-transcriptional control.

·       There are errors in the description of the Roquin family proteins, such as the claim that Roquin-1 and Roquin-2 don't have CC domains. Nevertheless, it is inconsistent to note in the paragraph that follows that Roquin-1 and Roquin-2 share a CC domain.

·       Information on how cytokine signalling and T cell receptor activation regulate Roquin proteins oversimplifies intricate regulatory systems. The evidence for MALT1's involvement in Roquin cleavage and inactivation is weak.The introduction's unclear structure makes it challenging for the reader to understand the ideas as they flow. It might be advantageous to divide the content up into separate paragraphs or sections according to themes or subjects.

Materials and methods:

·       Specific information on the parameters used in different analyses, like the alignment techniques, BLAST search settings, and motif analysis parameters, is missing from the approach. It is essential to provide these details in order to ensure reproducibility and comprehend the robustness of the analysis.

·       Although the usage of several tools and databases is mentioned in this section, there are no explanations or references to these resources. Clarity could be increased by including brief explanations or references for any tools mentioned that readers might not be acquainted with.

·       Although it's usual and handy to use internet tools for analysis, it's important to take into account their limitations and inherent biases. It would improve the process to include details on the validity and dependability of the instruments.

·       The reporting procedures used in this section are inconsistent between the various analyses. For instance, although some analyses are more generic, others offer step-by-step instructions. Clarity and readability would improve if the reporting format were standardised.

·       Based on a reference motif from mammals, the methodology mentions the discovery of putative binding motifs in zebrafish. It is imperative to recognise the possible constraints and variations in motif recognition and binding specificity among different species.

Results:

·       Although the results present a variety of conclusions, readers may find it difficult to completely understand the information due to the lack of accompanying visual aids like tables, figures, or diagrams. The results would be easier to understand and more accessible if they included visuals.

·       There is inconsistency in the presentation of the findings among the several subsections. For instance, although some subsections concentrate on qualitative descriptions, others offer comprehensive numerical data (such as molecular weights and expression levels). The organisation and readability of the results would both be enhanced by standardising the presenting format.

·       The findings are mostly described in the results, without going into further detail on their importance or wider ramifications. To enhance the analysis and facilitate readers' comprehension of the findings' importance, it would be beneficial to include further interpretation and discourse regarding the outcomes concerning the study questions or hypotheses.

·       Although the results note the use of Ka/Ks ratios as a gauge of selective pressure, they don't go into detail on the statistical techniques employed or the importance of the findings. Statistical analysis and significance testing details would improve the reliability of the results.

·       Insufficient evidence or context is provided for several of the sweeping generalisations made in the findings section. For example, the assertion that all species' Roquin genes descended from a single primordial Roquin-1 in invertebrates may oversimplify the evolutionary relationships if it ignores supporting data or other variables.

Discussion

·       The significance of RNA-binding proteins, like Roquin and Regnase, in mRNA degradation and immune system control is briefly reviewed before the topic is discussed. This lays the groundwork for appreciating the study's importance in clarifying the evolutionary and functional features of Roquin genes.

·       Effective comparisons between the new study's results and earlier studies are made throughout the discussion, especially with regard to the evolution and functional roles of the Regnase and Roquin genes. This comparison emphasises the contributions of the current study and places the new findings within the context of the body of existing literature.

·       The discussion offers a thorough examination of the evolution of Roquin genes, emphasising the effects of lineage-specific gene duplication/loss events and genome duplication events on the growth and diversification of the gene family. Our knowledge of the evolutionary dynamics of metazoan RNA-binding proteins has improved as a result of our analysis.

·       Roquin genes' putative functional activities are discussed, with a focus on their relevance to immunological responses, post-transcriptional control of mRNA, and genetic robustness maintenance. This talk highlights the importance of Roquin genes in biological processes and provides further context for the understanding of the data.

·       The study's conclusions and theories are ably supported by the discussion, which skillfully incorporates bioinformatic studies such as TF binding site predictions and Ka/Ks ratio computations. As a result, the findings made using the data are more credible.

·       Highlighting the study's key findings and their importance in expanding our knowledge of Roquin gene evolution and function, the debate comes to a close. The significance of the study and its possible ramifications for additional research in the area are emphasised by this.

Comments on the Quality of English Language

Extensive editing of English language required

Author Response

Response to Reviewer #1

Introduction:

The presentation of the introduction is not precise or well-defined. It lacks a clear transition between going into specifics of RNA binding proteins (RBPs) and the broader significance of controlling gene expression.

Reply: Thanks for the comments. We have included the following sentence for a smoother transition: “Gene expression is regulated through various mechanisms, such as epigenetic modifications, transcriptional regulation, and post-transcriptional regulation, which also involve multiple regulators.” This addition enhances the clarity between the specificity of RNA-binding proteins and gene regulation.

Certain statements, such "Abnormal gene expression leads to developmental defects and imbalanced immune responses," are not properly cited or supported by data. Although this is generally accurate, it seems like an unsupported claim in the absence of particular references.

Reply: Thank you for the comments. We have added the related references for “Abnormal gene expression leads to developmental defects and imbalanced immune responses,” to support our viewpoint.

It is overly generalised and simplistic to say that "RNA binding proteins play a crucial role in post-transcriptional control of gene expression, impacting almost all aspects of RNA metabolism". It ignores the variety of roles played by various RNA-binding proteins as well as the intricacy of post-transcriptional control.

Reply: Thanks a lot. We have rephrased the sentence to “RNA-binding proteins recognize specific RNAs through their RNA-binding domains, thereby regulating RNA splicing, localization, stability, and transport. The play crucial roles in various aspects including organ development, disease occurrence, and immune system homeostasis”.

There are errors in the description of the Roquin family proteins, such as the claim that Roquin-1 and Roquin-2 don't have CC domains. Nevertheless, it is inconsistent to note in the paragraph that follows that Roquin-1 and Roquin-2 share a CC domain.

Reply: We apologize for the confusion. Although description “human ROQUIN-2 lacks CC domain” was cited from [1], our analysis using SMART database prediction revealed the presence of a CC domain in human ROQUIN-2, and there was also other evidence for the presence of the CC domain in Roquin-2 [2], which may be due to different analytical methods and different algorithms that led to this result. We've removed “human ROQUIN-2 lacks CC domain” from Introduction.

Information on how cytokine signalling and T cell receptor activation regulate Roquin proteins oversimplifies intricate regulatory systems. The evidence for MALT1's involvement in Roquin cleavage and inactivation is weak. The introduction's unclear structure makes it challenging for the reader to understand the ideas as they flow. It might be advantageous to divide the content up into separate paragraphs or sections according to themes or subjects.

Reply: Thank you very much. We have included schemes in several sections of introduction. Furthermore, we have augmented the evidence supporting MALT cleaving ROQUIN and provided detailed explanations for this aspect.

Materials and methods:

Specific information on the parameters used in different analyses, like the alignment techniques, BLAST search settings, and motif analysis parameters, is missing from the approach. It is essential to provide these details in order to ensure reproducibility and comprehend the robustness of the analysis.

Reply: Thank you for the helpful comments. We have included detailed parameter information in the analyses, please see the revised text marked by yellow background.

Although the usage of several tools and databases is mentioned in this section, there are no explanations or references to these resources. Clarity could be increased by including brief explanations or references for any tools mentioned that readers might not be acquainted with.

Reply: Thank you for your suggestion. We have presented references to several tools and databases, include MEME tools, TBtools, Genomicus and PROMO database, highlighted with yellow background.

Although it's usual and handy to use internet tools for analysis, it's important to take into account their limitations and inherent biases. It would improve the process to include details on the validity and dependability of the instruments.

Reply: Thanks a lot. It is true that it is important of these databases and bioinformatics tools in enhancing our comprehension of evolutionary relationships, genetic structures, and more. However, it's also essential to recognize the limitations of these analysis. Nevertheless, the tools we employed in this study are widely utilized, and the overarching framework remains generally well accepted. Moving forward, we are committed to prioritizing the precision of these data in our future endeavors.

The reporting procedures used in this section are inconsistent between the various analyses. For instance, although some analyses are more generic, others offer step-by-step instructions. Clarity and readability would improve if the reporting format were standardised.

Reply: Thank you for your comments. The analyses conducted in this study encompass a range of well-established bioinformatics procedures, varying in complexity from straightforward to intricate. We adhere to standardized protocols for each analysis, which explains the inconsistencies observed among them.

Based on a reference motif from mammals, the methodology mentions the discovery of putative binding motifs in zebrafish. It is imperative to recognise the possible constraints and variations in motif recognition and binding specificity among different species.

Reply: Thank you so much. Recognition motifs and binding properties can vary from species to species. In mammals, Roquin exhibits a specific affinity for CDE motifs. Thus, we endeavored to predict putative CDE motifs in zebrafish. Our analysis revealed CDE-like motifs in numerous immune genes in zebrafish. Notably, these predicted CDE-like motifs demonstrate substantial homology with canonical CDE motifs. Consequently, we hypothesize that zebrafish Roquin may bind to CDE-like motifs within immune genes, similar to its mammalian counterpart.

Results:

Although the results present a variety of conclusions, readers may find it difficult to completely understand the information due to the lack of accompanying visual aids like tables, figures, or diagrams. The results would be easier to understand and more accessible if they included visuals.

Reply: Thank you. In order to present our results clearly, the manuscript includes 9 figures and 3 tables in the entire manuscript.

There is inconsistency in the presentation of the findings among the several subsections. For instance, although some subsections concentrate on qualitative descriptions, others offer comprehensive numerical data (such as molecular weights and expression levels). The organisation and readability of the results would both be enhanced by standardising the presenting format.

Reply: Thank you for the comments. All the analyses employed in this study aim to elucidate the evolution of Roquin and its potential role in the immune system. All the analyses conducted in this study are well established, and the results are presented in a standard way.

The findings are mostly described in the results, without going into further detail on their importance or wider ramifications. To enhance the analysis and facilitate readers' comprehension of the findings' importance, it would be beneficial to include further interpretation and discourse regarding the outcomes concerning the study questions or hypotheses.

Reply: Thanks a lot. We have elaborated on the results obtained and further discussed their implications. For details, please see the manuscript.

Although the results note the use of Ka/Ks ratios as a gauge of selective pressure, they don't go into detail on the statistical techniques employed or the importance of the findings. Statistical analysis and significance testing details would improve the reliability of the results.

Reply: Thanks for your suggestion. In the section of the Ka/Ks analysis, we have added the following statement: "In summary, the above analysis suggests that Roquin-1 and Roquin-2 are evolutionarily conserved and that their functions may be redundant to some extent." This statement aims to underscore the significance of Ka/Ks analysis in assessing evolutionary conservation and functional redundancy.

Insufficient evidence or context is provided for several of the sweeping generalisations made in the findings section. For example, the assertion that all species' Roquin genes descended from a single primordial Roquin-1 in invertebrates may oversimplify the evolutionary relationships if it ignores supporting data or other variables.

Reply: Thank you. We acknowledge that evolution is a complex process. It is evident that invertebrates possess only one Roquin gene, whereas most vertebrates have two. Notably, all the invertebrates observed in the evolutionary tree are located at the base of the vertebrate branch. Consequently, we hypothesize that Roquin across all species originated from a single primitive Roquin-1 present in invertebrates.

Discussion

The significance of RNA-binding proteins, like Roquin and Regnase, in mRNA degradation and immune system control is briefly reviewed before the topic is discussed. This lays the groundwork for appreciating the study's importance in clarifying the evolutionary and functional features of Roquin genes.

Effective comparisons between the new study's results and earlier studies are made throughout the discussion, especially with regard to the evolution and functional roles of the Regnase and Roquin genes. This comparison emphasises the contributions of the current study and places the new findings within the context of the body of existing literature.

The discussion offers a thorough examination of the evolution of Roquin genes, emphasising the effects of lineage-specific gene duplication/loss events and genome duplication events on the growth and diversification of the gene family. Our knowledge of the evolutionary dynamics of metazoan RNA-binding proteins has improved as a result of our analysis.

Roquin genes' putative functional activities are discussed, with a focus on their relevance to immunological responses, post-transcriptional control of mRNA, and genetic robustness maintenance. This talk highlights the importance of Roquin genes in biological processes and provides further context for the understanding of the data.

The study's conclusions and theories are ably supported by the discussion, which skillfully incorporates bioinformatic studies such as TF binding site predictions and Ka/Ks ratio computations. As a result, the findings made using the data are more credible.

Highlighting the study's key findings and their importance in expanding our knowledge of Roquin gene evolution and function, the debate comes to a close. The significance of the study and its possible ramifications for additional research in the area are emphasised by this.

Reply: We appreciated your helpful comments. The suggestions are helpful and will significantly improve this work and our future work. Thank you sincerely.

References

  1. Athanasopoulos V, Ramiscal RR, Vinuesa CG. ROQUIN signalling pathways in innate and adaptive immunity. Eur J Immunol. 2016;46:1082-90.
  2. Sakurai S, Ohto U, Shimizu T. Structure of human Roquin-2 and its complex with constitutive-decay element RNA. Acta Crystallogr F Struct Biol Commun. 2015;71:1048-54.

Reviewer 2 Report

Comments and Suggestions for Authors

The manuscript by Li et al. focuses on bioinformatics analysis of the Roquin family. Based on the analysis, the authors claimed that RC3H genes underwent a single round of gene duplication during its evolution from invertebrate to vertebrate. Some species-specific gene lost events and teleost lineage-specific gene duplications were also identified. The authors put effort into this work, and the calculations were performed in a detailed manner. The paper is well-written and, in my opinion, presented results are a good starting point for experimental (in vitro) study, e.g., gene(s) knockouts. Therefore, the paper is a strong candidate for this journal if the issues below are adequately addressed in a revised version.

1.      It would be convenient to include some schemes related to the introduction section, e.g., for structural and functional aspects of the ROQUIN family.

2.      The quality of the figures has to be improved. For example, the fonts and figures’ resolution are too small. Bootstrap values should be included on the trees presented in Figure 1.

3.      Some English language corrections (tenses, grammar, etc.) should be applied throughout the text.

Comments on the Quality of English Language

Some English language corrections (tenses, grammar, etc.) should be applied throughout the text.

Author Response

Response to Reviewer #2

  1. It would be convenient to include some schemes related to the introduction section, e.g., for structural and functional aspects of the ROQUIN family.

Reply: Thanks for your suggestions. We have incorporated suitable schemes for various section of the Introduction, see the main text, please.

  1. The quality of the figures has to be improved. For example, the fonts and figures’ resolution are too small. Bootstrap values should be included on the trees presented in Figure 1.

Reply: Thank you. We have modified the figures according to your suggestions. See the revised figures, please.

  1. Some English language corrections (tenses, grammar, etc.) should be applied throughout the text.

Reply: We apologize for the poor language of our manuscript. The language of our manuscript has been modified by the suggested English Editing services (MDPI).
